# Why are hyperbolic neural networks effective? A study on hierarchical representation capability

## Abstract

Hyperbolic Neural Networks (HNNs), operating in hyperbolic space, have been widely applied in recent years, motivated by the existence of an optimal embedding in hyperbolic space that can preserve data hierarchical relationships (termed Hierarchical Representation Capability, HRC) more accurately than Euclidean space. However, there is no evidence to suggest that HNNs can achieve this theoretical optimal embedding, leading to much research being built on flawed motivations. In this paper, we propose a benchmark for evaluating HRC and conduct a comprehensive analysis of why HNNs are effective through large-scale experiments. Inspired by the analysis results, we propose several pre-training strategies to enhance HRC and improve the performance of downstream tasks, further validating the reliability of the analysis. Experiments show that HNNs cannot achieve the theoretical optimal embedding. The HRC is significantly affected by the optimization objectives and hierarchical structures, and enhancing HRC through pre-training strategies can significantly improve the performance of HNNs. [1]

## 1 Introduction

Exploiting the unique advantages of hyperbolic space, **H**yperbolic **N**eural **N**etworks (**HNN**s) have revolutionized the handling of massive hierarchical data (Sala et al., 2018; Chami et al., 2020; Cao et al., 2020; Nickel & Kiela, 2017), providing a more accurate representation (referred to as **H**ierarchical **R**epresentation **C**apability, **HRC**) than traditional Euclidean-based methods (Ganea et al., 2018).

However, existing research on hyperbolic space performance only proves the minimum distortion of embedding in hyperbolic space in theory (Sala et al., 2018; Tabaghi & Dokmanić, 2020) and does not prove that any method used in hyperbolic space has the best HRC. Suzuki et al. (2021) theoretically demonstrated that the effectiveness of hyperbolic space is only limited to ideal noiseless settings, and less data and imbalanced data distribution may worsen errors. Especially for specific HNN methods, their performance will obviously be affected by optimization objectives and data. Agibetov et al. (2019) has noticed the phenomenon that classifiers in hyperbolic spaces are inferior to Euclidean spaces.

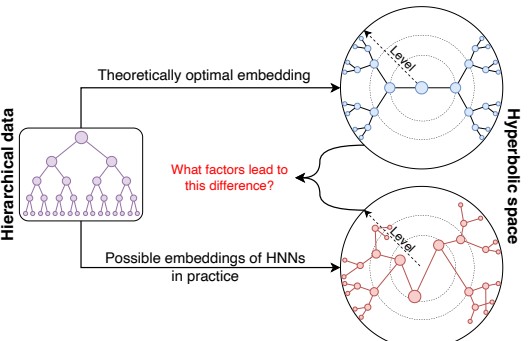

Figure 1: In theory, there exists an optimal embedding for hierarchical data in hyperbolic space, but HNNs can be affected by various factors and may not necessarily achieve the optimal embedding. Therefore, the effectiveness of HNNs cannot simply be attributed to the HRC of hyperbolic spaces.

Referring to Figure 1, in order to elucidate the factors influencing HRC, we propose a **H**ierarchical **R**epresentation **C**apability **B**enchmark (**HRCB**) for evaluating the HRC. HRCB designs evaluation

---

[1]The code is available at `https://anonymous.4open.science/r/HRCB`

metrics based on the distance relationship between parent-child nodes and those between sibling nodes in the tree structure, as well as between these nodes and the root node and hyperbolic space origin, in order to quantitatively analyze the impact of HRC under different factors.

The existing methods overlook the impact of HRC on performance, whereas we aim to improve HNNs by taking HRC as our guiding principle. Based on the conclusions obtained from using HRCB for analysis, we propose various pre-training strategies to enhance HRC, in order to improve the performance of HNNs and verify the correctness of the analysis.

In summary, our major contributions are:

- We propose HRCB to evaluate the HRC of HNNs. The HRCB will enable researchers to assess the scope and applicability of HNNs and gain insights into their underlying mechanisms. Our analysis of HNNs has uncovered specific factors contributing to their effectiveness.
- We propose various pre-training strategies to enhance HRC and analyze the relationship between HRC and downstream task performance. Our proposed strategy further validates and leverages the analytical results on HRCB to improve the performance of HNNs within the applicable scope.
- To ensure the reliability of our analysis and improvements, we conducted thousands of experiments on three model structures, three manifold spaces, and eight dimensions. To analyze the extensive experimental results, we used statistical significance tests to verify the conclusions.

## 2   RELATED WORK

The motivation of HNNs is derived from the study of the properties of hyperbolic space. Gromov (1987); Linial et al. (1995) theoretically demonstrated the superiority of hyperbolic space for the tree structure representation (Law et al., 2019). Subsequent extensive research (Bonahon, 2009; Krioukov et al., 2010; Sarkar, 2011; Sala et al., 2018; Tabaghi & Dokmanić, 2020; Sonthalia & Gilbert, 2020) demonstrated that embedding in hyperbolic spaces with minimal hierarchical distortion is possible and provided embedding methods for known trees.

The study of the properties of hyperbolic space inspired the rapid development of HNNs. Ganea et al. (2018) firstly proposed neural network operations on Poincaré ball model (an analytic model of hyperbolic space). Nickel & Kiela (2018) proposed neural network operations on Hyperboloid model. Since then, a large number of studies transferred neural network models to hyperbolic space, such as graph convolutional networks (Chami et al., 2019; Yang et al., 2022a; Sun et al., 2021; Shimizu et al., 2021), neural networks for image processing (Skopek et al., 2020; Lazcano et al., 2021; Zhang et al., 2022; Ahmad & Lécué, 2022), neural networks for text analysis (Chen et al., 2020; Zhu et al., 2020; Chen et al., 2021; Agarwal et al., 2022; Song et al., 2022), deep reinforcement learning (Cetin et al., 2023), and so on. The aforementioned methods assert that the performance of HNNs originates from HRC, however, no investigation has been conducted to explore the relationship between HRC and performance.

Some work has discussed how to measure the HRC. Gu et al. (2019) used graph distortion and MAP to show graph reconstruction performance, but they cannot accurately describe the hierarchical relationships between nodes before reaching theoretical optimality. Furthermore, some visualization methods (Nickel & Kiela, 2017; Chami et al., 2019; Mathieu et al., 2019) make it difficult to quantitatively assess HRC.

Although a considerable amount of work has analyzed the properties and advantages of hyperbolic space, little research has focused on the properties and advantages of HNNs, instead referring to the analysis of hyperbolic space (Weber et al., 2020; Yang et al., 2022b). Therefore, we elucidate why HNNs are effective through the proposed HRCB and, based on the analysis results, propose pre-training strategies to enhance the performance of HNNs.

## 3   BENCHMARK DESIGN

In this section, we introduce two components of HRCB: the evaluation metrics for HRC and the description and generation of hierarchical structures, to analyze the impact of different optimization objectives and hierarchical structures on the HRC.

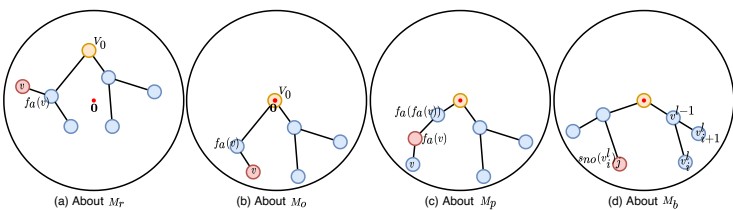

Figure 2: Four examples to help improve the four evaluation metrics ($M_r$, $M_o$, $M_p$, $M_b$). The range of the four evaluation metrics is between 0 and 1, the larger the better.

## 3.1 EVALUATION METRICS

To address the challenge that HRC is difficult to evaluate, we propose four distinct evaluation metrics. These metrics are based on the parent and sibling nodes within the hierarchy, along with their distance relationships to both the root node and the origin of the hyperbolic space. Inspired by the galaxy structure (Du et al., 2018), the hierarchical tree should preserve the horizontal (with respect to sibling nodes) and the vertical (with respect to parent-child nodes) relationships. We extend this idea to the evaluation of hierarchical structures in hyperbolic space. Specifically, we design four evaluation metrics:

**Root Node Hierarchy Metric** ($M_r$). A node should be further away from the root than its parent. As shown in Figure 2(a), if node $v$ is further away from the root node $V_0$ than its parent node $f_a(v)$, then this embedding of the hierarchical structure is consistent with the tendency of the volume to increase exponentially with radius in hyperbolic space. We can use a metric $M_r$ to show the proportion of nodes satisfying Figure 2(a) to all nodes.

**Coordinate Origin Hierarchy Metric** ($M_o$). A node should be further away from the coordinate origin than its parent. As shown in Figure 2(b), the node $v$ is further away from the origin $\mathbf{0}$ than its parent $f_a(v)$, because some operations (such as aggregation, activation functions, etc.) need to be performed on the tangent space of the origin, and a hierarchical structure around the origin makes more sense. We can use a metric $M_o$ to show the proportion of nodes satisfying Figure 2(b) to all nodes.

**Parent Node Hierarchy Metric** ($M_p$). A node should be further away from the grandfather node than its parent. As shown in Figure 2(c), if a node $v$ is further away from its grandfather node $f_a(f_a(v))$ than its parent node $f_a(v)$, then this embedding is consistent with the parent-child node relationship in the hierarchical structure. We can use a metric $M_p$ to show the proportion of nodes satisfying Figure 2(c) to all nodes.

**Sibling Node Hierarchy Metric** ($M_b$). The distance between sibling nodes should be smaller than the distance between non-sibling (and non-parent-child) nodes. As shown in Figure 2(d), if $d_{\mathcal{M}}(v_i^l, v_{i+1}^l) < d_{\mathcal{M}}(v_i^l, s_{no}(v_i^l)_j)$, then this embedding is consistent with the sibling node relationship in the hierarchical structure. By hierarchical traversal, we can use a metric $M_b$ to show the proportion of distance relations satisfying Figure 2(d) to all distance relations.

The formal descriptions of $M_r$, $M_o$, $M_p$, and $M_b$ can be found in Appendix A.1.

## 3.2 HIERARCHICAL STRUCTURES

Since embedding for tree (hierarchical) structures can maximize the effectiveness of hyperbolic space (Sala et al., 2018; Chami et al., 2019), HRCB evaluates using a tree structure rather than a tree-like hierarchical structure. Different hierarchical structures are challenging to describe and construct. The existing works treat all hierarchical structures equally and do not have a deeper analysis. To address the challenge of describing different hierarchical structures, we propose two metrics that focus on the distribution characteristics of the nodes to effectively describe the hierarchical structure. Inspired by the balanced tree, we measure the change of subtree height and degree distribution of nodes in horizontal and vertical directions:

**Horizontal Hierarchical Difference** ($I_B$). The height difference of subtrees between sibling nodes. Some higher subtrees may affect the number of nodes to increase exponentially with the nodes

hierarchy, so we need to analyze this type of hierarchical structure. We measure the height difference by the average standard deviation of subtree heights (See Figure 3 (a) and (b) for what it means).

**Vertical Degree Distribution** ($I_D$). As the node hierarchy is higher, does the degree of nodes also increase? This change may affect the tendency of the number of nodes to increase exponentially with the nodes hierarchy, so we need to analyze this type of hierarchical structure. This is similar to the evaluation of ranking algorithms, so we borrowed the normalized discounted cumulative gain (NDCG) to measure this change (See Figure 3 (c) and (d) for what it means).

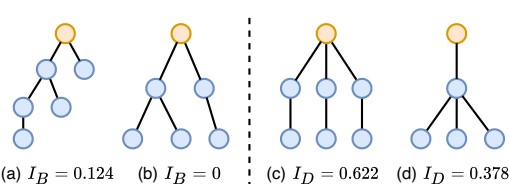

(a) $I_B = 0.124$  (b) $I_B = 0$  (c) $I_D = 0.622$  (d) $I_D = 0.378$

Figure 3: Examples of horizontal hierarchical difference ($I_B$): it is clear that Figure (a) is more "unbalanced" than Figure (b), so the $I_B$ of Figure (a) is larger. Examples of vertical degree distribution ($I_D$): the closer to the root node in Figure (c), the greater the degree, so $I_D > 0.5$, and vice versa, as in Figure (d), $I_D < 0.5$.

Real-world data struggles to cover all types of hierarchical structures, necessitating a rational approach to generate hierarchies with diverse characteristics. Specifically, we control the probability of generating child nodes from left to right and the number of child nodes from top to bottom, thus obtaining the hierarchical structure with varying $I_B$ and $I_D$, respectively.

The formal descriptions of $I_B$, $I_D$, and the generation of hierarchical structures can be found in Appendix A.3.

## 4 HRC ENHANCEMENT

In addition to analyzing why HNNs are effective, we also aim to further improve HRC within their applicable scope to enhance downstream task performance and investigate whether these capabilities impact the performance of downstream tasks. To achieve this, we propose three pre-training strategies using optimization objectives beneficial to HRC as pre-training targets, as illustrated in Figure 4. We first enhance the encoder's HRC using a pre-training target, then apply the encoder to downstream tasks. Theoretically, pre-training target can be any objective other than the downstream task targets, which will be further discussed in the experimental section.

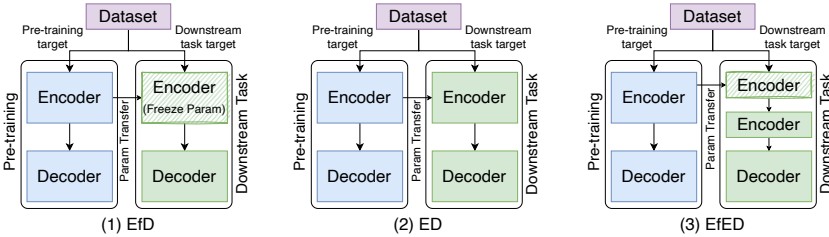

Figure 4: Three pre-training strategies for enhancing HRC. (1) EfD: Apply the HRC-enhanced encoder directly to downstream tasks while freezing its parameters. (2) ED: Apply the HRC-enhanced encoder directly to downstream tasks without freezing its parameters. (3) EfED: Place the HRC-enhanced encoder before the downstream task's encoder and freeze its parameters.

## 5 EXPERIMENTS

In this section, we show that the HRC of the HNNs is significantly lower than the upper limit of the hyperbolic space. We also show that the optimization objective that helps distinguish the position relationship between any two nodes and the hierarchical structure that approximates a complete binary (or N-ary) tree can help to improve HRC. Specifically, we aim to answer the following three key questions.

- Can HNNs' HRC reach the upper limit of hyperbolic space? In other words, can HNNs achieve optimal embeddings in hyperbolic space?

- What factors influence HNNs' HRC? Alternatively, how does HNNs' HRC get affected by optimization objectives and hierarchical structures?
- Does HNNs' HRC impact the performance of downstream tasks? In other words, can improving HRC lead to enhanced performance in downstream tasks?

These questions can help us understand why HNNs are effective, shedding light on the motivation and applicability of HNNs.

## 5.1 EXPERIMENTAL SETUP

Our experiments were conducted on the TensorFlow framework, utilizing both the NVIDIA Tesla V100 and NVIDIA RTX 3090 for training.

**Datasets**. In addition to datasets with all the hierarchical structures in Section 3.2, we use two public hierarchical datasets for the analysis of optimization objectives and embedding methods (HNNs): Disease (Chami et al., 2019) and WordNet (Miller, 1998). Since WordNet is too computationally demanding, we use the usual Animal (Shimizu et al., 2021) subset of it. Since WordNet is not a tree, we only kept the longest path from the node to the root node. See Appendix B.1 for more details.

**Manifold Spaces**.

In order to comprehensively analyze the HRC in hyperbolic space, we employ the two most representative and commonly used analytic models in hyperbolic space (Peng et al., 2021): the Poincaré ball model $\mathbb{D}$ and the Hyperboloid model $\mathbb{H}$. Together with Euclidean space $\mathbb{R}$, they constitute the three manifold spaces analyzed in this paper. See Appendix B.2 for more details.

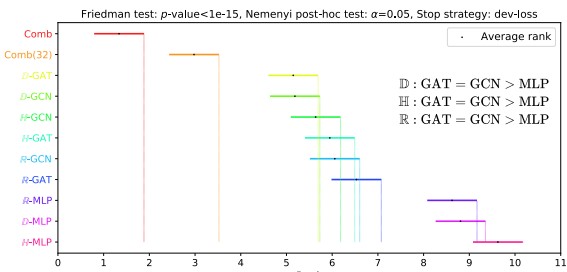

**Hyperbolic Neural Networks**. HNNs aim to embed objects such as images (Khrulkov et al., 2020), texts (Zhang & Gao, 2020), and networks (Wang et al., 2021), among which network embeddings are the most suitable for studying HRC due to their more intuitive hierarchical structure. We employed the most commonly used and well-established network embedding models in hyperbolic space, including MLP (Ganea et al., 2018), GCN (Liu et al., 2019), and GAT (Zhang et al., 2019). See Appendix B.3 for more details.

**Optimization Objectives**. Optimization objectives play a crucial role in determining the quality and effectiveness of node embeddings in HNNs, directly influencing

Figure 5: Friedman test and Nemenyi post-hoc test for eleven methods ($\{\mathbb{R}, \mathbb{D}, \mathbb{H}\} \times$ {GAT,GCN,MLP}+Comb+Comb(32) with 32-bit floating point precision). Comb has a floating point precision of 3000 bits, while other HNN methods have a floating point precision of 32 bits. Significance level $\alpha$ is 0.05. We also performed significance tests for all evaluation metrics ($M_r, M_o, M_p, M_b$) related to the HRC, so that each method contains 192 experimental results (eight dimensions $\times$ {GD,HR,FD} $\times$ {Animal,Disease} $\times$ {$M_r, M_o, M_p, M_b$}). We removed the LR (Logistic Regression, NC tasks) unrelated to HRC, and the rationale behind this decision is discussed in Section 5.2.2. Where > denotes significantly better than, = denotes not significantly better than.

their overall performance. We introduce optimization objectives commonly used in downstream tasks for graph representation learning. The objectives include Graph Distortion (GD), Hypernymy Relations (HR), Fermi-Dirac with cross-entropy (FD), and Logistic Regression (LR), while the tasks are Graph Reconstruction (GR), Link Prediction (LP), and Node Classification (NC). See Appendix B.4 for more details.

**Dataset Splitting Strategy**. The train/dev/test ratio of the dataset used for the NC task (corresponding to LR) is 3:1:6. The train/dev/test ratio of the dataset used for the other tasks is 8:1:1.

**Hyperparameter**. The model has no hyperparameters that need to be optimized. By convention, the encoder is set as a two-layer neural network, the GAT uses 4-head attention, the activation function uses ReLU, the optimizer is Riemann Adam, and the learning rate is 0.01. For the encoder output layer dimensions, we experimented with eight dimensions (2,4,6,8,10,12,14,16) on each result.

**Stop Strategy**. Our stop strategy is set to stop training when the value of the loss function in the validation set stops decreasing within 100 epochs. In addition, we also keep the result of using the loss function of the training set as the stop strategy for more in-depth analysis.

## 5.2 RESULTS AND ANALYSIS

### 5.2.1 CAN HNNS' HRC REACH THE UPPER LIMIT OF HYPERBOLIC SPACE?

To answer this question, we compare the HRC for different embedding methods on different manifold spaces and datasets. In addition to MLP, GCN, and GAT, we also include a method that does not require optimization objectives and neural networks: Combinatorial Constructions (Comb) (Sala et al., 2018). This method is to place the nodes into hyperbolic space ($\mathbb{D}$) with the known hierarchical structure, which can almost reach the upper limit of the HRC. Our goal is to utilize the strength of Comb to critically analyze the gap between other methods and the optimal embedding.

To obtain more accurate conclusions, we show the significance tests for all embedding methods on different manifold spaces in the form of Friedman test charts in Figure 5. **The HRC of HNNs is significantly lower than the upper limit of hyperbolic space.** We can see that the ranking of the methods is Comb > Comb(32) > GAT = GCN > MLP. All the hyperbolic neural network methods are significantly weaker than Comb(32). The main reason is that these methods are only transferred to hyperbolic space, without improvements for the hierarchical structure. Therefore, besides transferring the methods to hyperbolic space, it is worth exploring how to improve the neural network methods to suit the hierarchical structure.

### 5.2.2 WHAT FACTORS INFLUENCE HNNS' HRC?

To address this question, we first compared the HRC of four optimization objectives across three manifold spaces and two datasets. We show the significance tests for all optimization objectives on different manifold spaces in the form of Friedman test charts in Figure 6. From Figure 6(a), it can be seen that with the standard stop strategy (dev-loss), the ranking of the optimization objectives is Graph Distortion (GD) $\geqslant$ Hypernymy Relations (HR) > Fermi-Dirac with cross-entropy (FD) > Logistic Regression (LR). Different optimization objectives have significant effects on HRC, and we have the following observations and analysis.

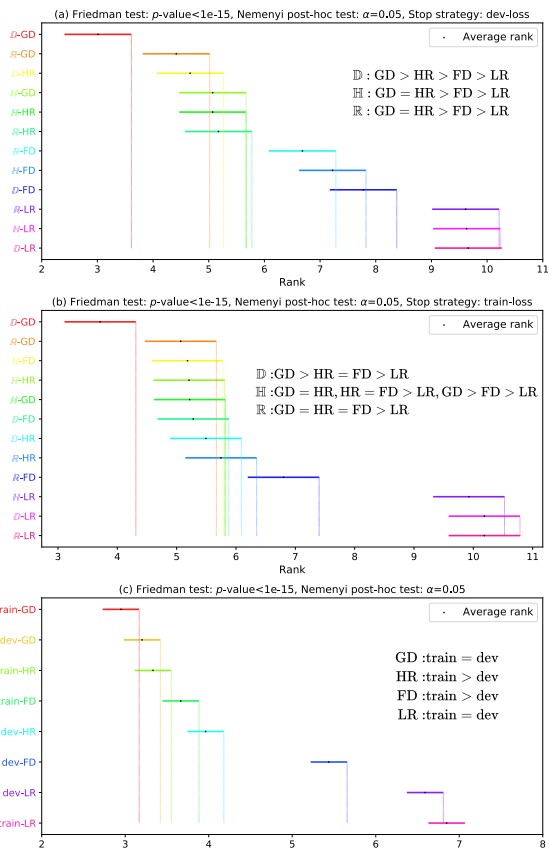

Figure 6: (a) and (b): Friedman test and Nemenyi post-hoc test for twelve methods ($\{\mathbb{R}, \mathbb{D}, \mathbb{H}\} \times \{GD,HR,FD,LR\}$). Significance level $\alpha$ is 0.05. We also performed significance tests for all evaluation metrics ($M_r, M_o, M_p, M_b$) related to the HRC, so that each method contains 192 experimental results (eight dimensions $\times\{MLP,GCN,GAT\} \times \{Animal,Disease\} \times \{M_r, M_o, M_p, M_b\}$). To analyze the impact of overfitting training on the HRC, we record the results of both stop strategies: the validation set loss function values in Figure (a), and the training set loss function values in Figure (b). Figure (c): Comparison of eight methods (two stop strategies $\times\{GD,HR,FD,LR\}$), each method contains 576 experimental results ($\{\mathbb{R}, \mathbb{D}, \mathbb{H}\}\times 192$). In this figure, > denotes significantly better than, = denotes not significantly better than.

(1) **Optimization objectives that help to distinguish the position relation between any two nodes also help to improve the HRC.** The shortest path of GD, hypernymy relations of HR, and edge distance of FD help the hyperbolic neural network method to distinguish this position relation. For nodes $v_a, v_b, v_c$, suppose there exists a true position relation $d_{\mathcal{M}}(v_a, v_b) < d_{\mathcal{M}}(v_a, v_c)$, if the loss function is equal to zero while there exists $d_{\mathcal{M}}(v_a, v_b) > d_{\mathcal{M}}(v_a, v_c)$, then this is the phenomenon that does not help to distinguish the position relation between any two nodes. The LR used for NC task does not need to distinguish whether the nodes of the same class belong to different hierarchical structures, so this phenomenon exists in abundance. Therefore, only considering the hierarchical structure is not enough, and optimization objectives or downstream tasks are also very important.

(2) **For optimization objectives that help to distinguish the position relation between any two nodes, overfitting may improve the HRC.** From Figure 6(b), it can be seen that the ranking of the optimization objectives is GD $\geqslant$ HR $\geqslant$ FD > LR under the stop strategy (train-loss) that allows overfitting. From Figure 6(c), we can further see that the HRC of HR and LR improves significantly when overfitting on the training set. This is because improving HRC through overfitting does not necessarily translate to better performance in downstream tasks.

To further address how different hierarchical structures impact HNNs' HRC, we show the trends of the five evaluation metrics with $I_B$ on horizontal hierarchical difference and $I_D$ on vertical degree distribution, as shown in Figure 7. Different hierarchical structures have significant effects on HRC. **The more the hierarchical structure approximates the complete n-ary tree, the more it helps to improve the HRC.** Most of the evaluation metrics show this trend, for example, $M_r$ and $M_o$ decreases and $M_{dd}$ (normalized graph distortion) increases when $I_B > 0$, $M_b$ and $M_p$ decreases when $I_D < 0.5$, and $M_b$ decreases when $I_D > 0.5$. With one exception, $M_p$ increases when $I_B > 0, I_D > 0.5$. Because there are fewer nodes at the same level in this case, the parent nodes between the grandfather and child nodes are not easily misaligned. In general, $I_B > 0$ and $I_D \neq 0.5$ will affect the exponential increase of the number of nodes with the hierarchy, so the advantage of hyperbolic space cannot be fully exploited.

In graph structures, a node may also belong to multiple hierarchical structures. To analyze the HRC of the HNNs in this case, we compare Mix-tree (a blend of multiple hierarchical structures) with Sub-tree (a single hierarchical structure). Figure 8 shows the results of the significance test. **If a node belongs to multiple hierarchical structures at the same time, the HRC of HNNs decreases significantly.** It can be seen that the Sub-tree significantly outperforms the Mix-tree, except for the unbalanced tree (T2/T6). Since a node has only one embedding, it is difficult to distinguish multiple hierarchical structures.

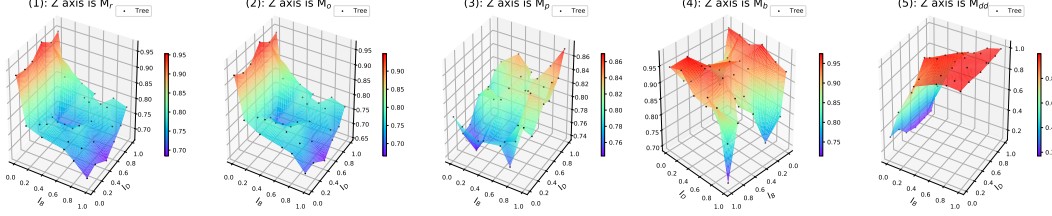

Figure 7: 36 datasets with $I_B, I_D$, and their results on five evaluation metrics ($M_r, M_o, M_p, M_b, M_{dd}$). We obtained 36 hierarchical structures with different $I_B$ and $I_D$ using the hill-climbing algorithm. These datasets are approximately uniformly distributed over the range of values of $I_B$ and $I_D$, and each dataset contains 3280 nodes. Each result for each dataset (Tree) is the average of 48 sets of experiments ($\{\mathbb{D}, \mathbb{H}\} \times \{\text{GD,HR,FD}\} \times$ eight dimensions$\times$GCN). $M_{dd}$ denotes normalized graph distortion.

### 5.2.3 DOES HNNS' HRC IMPACT THE PERFORMANCE OF DOWNSTREAM TASKS?

To answer this question, we compared the HRC and performance of strategies with and without HRC enhancement across four downstream task targets. As illustrated in Figure 9's Friedman test charts, the four downstream task targets include Logistic Regression (LR), Fermi-Dirac with cross-entropy (FD), Graph Distortion (GD), and Hypernymy Relations (HR). For LR, FD, and HR, the performance metric is accuracy, while for GD, the performance metric is the negative normalized distortion degree ($-M_{dd}$). In addition to the Normal strategy without HRC enhancement and the three HRC-enhanced pre-training strategies (ED, EfD, EfED), we also compared an HRC enhancement strategy called

L, which trains the model by weighting and summing the pre-training and downstream task targets. L0.5, L0.6, L0.7, L0.8, and L0.9 represent the HRC-enhanced L strategy with downstream task target weights of 0.5, 0.6, 0.7, 0.8, and 0.9, respectively. It is worth noting that the HRC enhancement strategy for the NC task has three pre-training targets (FD, GD, HR) to choose from, while the HRC enhancement strategy for other tasks only has two pre-training targets available, as the conclusion in Section 5.2.2 has confirmed that the optimization objective (i.e., LR) for the NC task is not suitable for enhancing HRC.

Apart from the NC task (targeted at LR), Figure 9 shows that most of the strategies capable of enhancing HRC also outperform the Normal strategy in terms of performance. For instance, out of the 17 strategies that significantly outperform Normal in HRC, only three (such as ED, L0.5, and L0.9 in Figure 9(h)) do not exhibit a significant improvement in performance. Enhancing HRC on the LR target, however, leads to reduced performance, as demonstrated by the seven strategies with significantly improved HRC in Figure 9(a) having notably lower performance in Figure 9(b) compared to the Normal strategy. **This indicates that the HRC of HNNs has a significant impact on the performance of downstream tasks.** Within the applicable scope of HNNs, performance can be improved by enhancing HRC, while outside this scope, increased HRC may lead to decreased performance. This also validates the effectiveness of the proposed pre-training strategies.

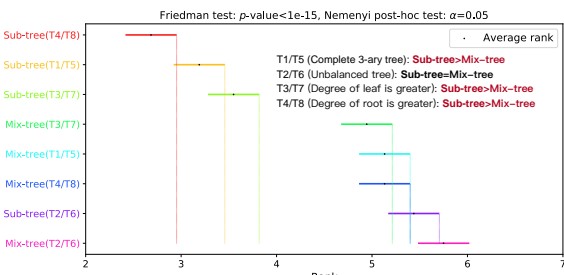

Figure 8: Friedman test and Nemenyi post-hoc test for eight methods ({Mix-tree,Sub-tree} × {T1/T5,T2/T6,T3/T7,T4/T8}). Significance level $\alpha$ is 0.05. Each method contains 384 experimental results ({Ti,Tj}× {$M_r, M_o, M_p, M_b$} × {$\mathbb{D}, \mathbb{H}$} × {GD,HR,FD}× eight dimensions ×GCN). Mix-tree denotes training with Tree5 (mixed by T1-T8) first, and then evaluating T1-T8 separately. Sub-tree denotes that T1-T8 are trained and evaluated separately. Appendix A.3 provides further details on Tree5 and T1-T8.

After verifying the impact of HNNs' HRC on downstream task performance, we analyzed the most optimal HRC enhancement strategy. As seen in Figure 9, we concluded that: **(1) Without considering the applicable scope of HNNs, ED is the best among various pre-training strategies.** ED's performance is not significantly weaker than the Normal strategy, and in FD and GD targets, ED outperforms the Normal strategy significantly (see Figure 9(d) and Figure 9(f)). This is attributed to ED not fixing parameters, which offers greater flexibility in retaining or forgetting pre-training targets and more effectively forgetting pre-training target even in unsuitable NC tasks for HNNs. **(2) Within the applicable scope of HNNs, EfD is the most optimal among various pre-training strategies.** In the three downstream task targets suitable for HNNs, EfD achieves the best HRC and performance on two targets (see Figure 9(d) and Figure 9(h)). This is because EfD fixes the Encoder's parameters, maximizing the preservation of the pre-training target's advantages. **(3) GD is more suitable as a pre-training target for pre-training strategies.** Among the four downstream task targets, only GD exhibits a phenomenon where the HRC of seven HRC-enhanced strategies does not improve (see Figure 9(e)). This is because GD cannot use GD as a pre-training target, and other pre-training targets are not as effective as GD in terms of HRC enhancement.

## 6 CONCLUSION

To understand why hyperbolic neural networks (HNNs) are effective, we propose a benchmark (HRCB) for quantitatively analyzing the hierarchical representation capability (HRC) of HNNs. We discover that the effectiveness of HNNs stems from two aspects: (1) HNNs' optimization objectives facilitate distinguishing positional relationships between any two nodes, and (2) HNNs' training data approximates a complete n-ary tree. This further clarifies the motivation and applicability of HNNs, validating that effective HNNs are not solely due to the hierarchical structure of the data. Based on HRCB's analysis of HNNs' motivation, we propose various pre-training strategies to further enhance HNNs' HRC, thereby improving their performance on downstream tasks. Experimental results indicate that HNNs' HRC significantly impacts downstream task performance, and enhancing HRC through pre-training strategies can substantially boost HNNs' performance.

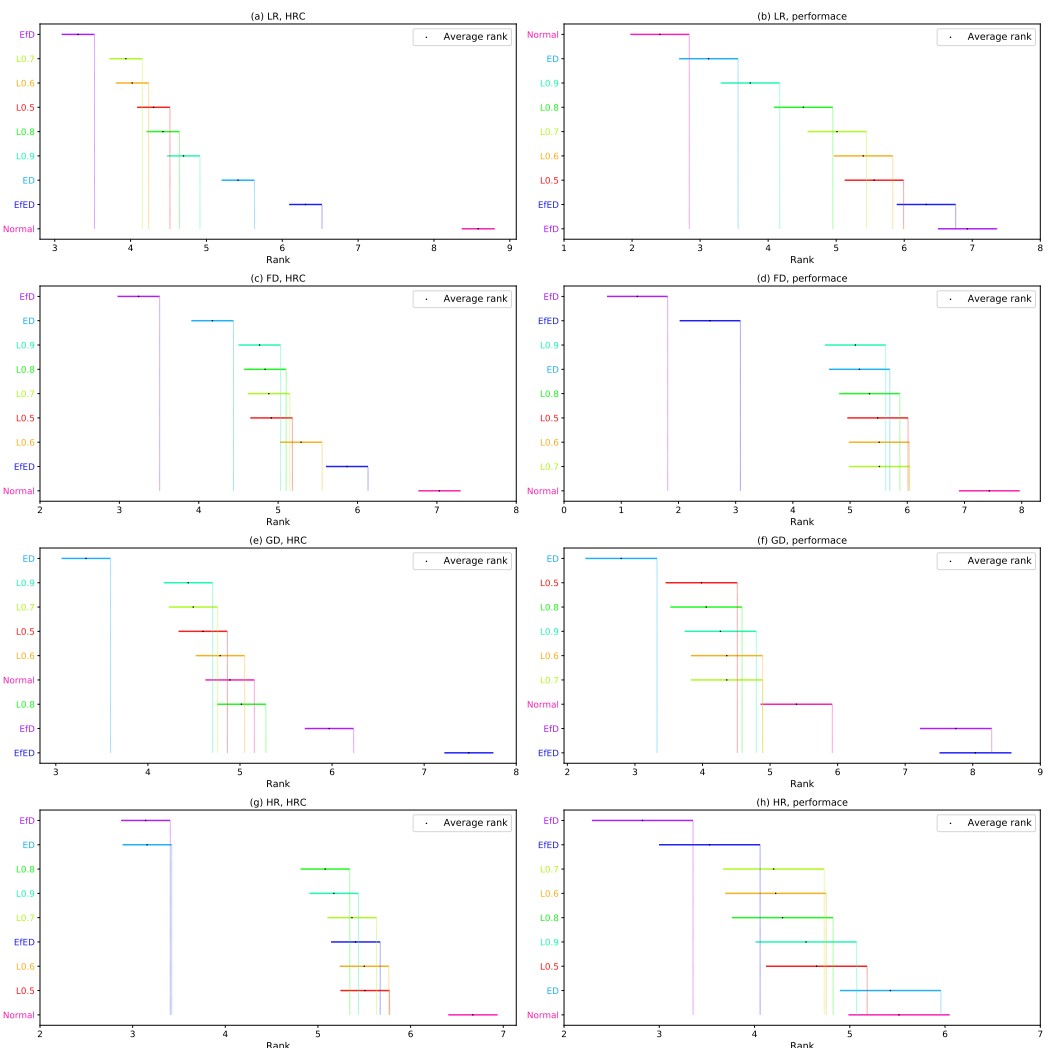

Figure 9: Friedman test and Nemenyi post-hoc test for nine strategies on four downstream task targets. Significance level $\alpha$ is 0.05. We performed significance tests for all evaluation metrics $(M_r, M_o, M_p, M_b)$ related to the HRC. For the downstream task target LR, there are three available pre-training targets (FD, GD, HR). Thus, the eight HRC enhancement strategies in Figure (a) each contain 768 experimental results (three pre-training targets $\times$ eight dimensions $\times$ $\{\mathbb{D}, \mathbb{H}\}$ $\times$ {GCN,GAT} $\times$ {Animal,Disease} $\times$ $\{M_r, M_o, M_p, M_b\}$), while those in Figure (b) each contain 192 experimental results (three pre-training targets $\times$ eight dimensions $\times$ $\{\mathbb{D}, \mathbb{H}\}$ $\times$ {GCN,GAT} $\times$ {Animal,Disease} $\times$ {Accuracy}). For the other three downstream task targets (FD, GD, HR), there are only two available pre-training targets. Consequently, the eight HRC enhancement strategies in Figure (c), Figure (e), and Figure (g) each contain 512 experimental results (two pre-training targets $\times$ eight dimensions $\times$ $\{\mathbb{D}, \mathbb{H}\}$ $\times$ {GCN,GAT} $\times$ {Animal,Disease} $\times$ $\{M_r, M_o, M_p, M_b\}$), and those in Figure (d), Figure (f), and Figure (h) each contain 128 experimental results (two pre-training targets $\times$ eight dimensions $\times$ $\{\mathbb{D}, \mathbb{H}\}$ $\times$ {GCN,GAT} $\times$ {Animal,Disease} $\times$ {Accuracy}). The Normal strategy does not include pre-training targets, so the Normal strategy on HRC comprises 256 experimental results (eight dimensions $\times$ $\{\mathbb{D}, \mathbb{H}\}$ $\times$ {GCN,GAT} $\times$ {Animal,Disease} $\times$ $\{M_r, M_o, M_p, M_b\}$), and the Normal strategy on performance contains 64 experimental results (eight dimensions $\times$ $\{\mathbb{D}, \mathbb{H}\}$ $\times$ {GCN,GAT} $\times$ {Animal,Disease} $\times$ {Accuracy}). The Normal strategy results need to be duplicated two or three times to match the results of the other eight strategies, facilitating significance testing.

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

## A    HRCB DETAILS

In this section, we provide a formal description and an in-depth analysis of the metrics proposed in Section 3. Table 1 lists the main notations used in this section.

Table 1: Notations.

| Symbol | Description |
|--------|-------------|
| $\mathbf{V}$ | A set of all nodes on a hierarchical structure |
| $V_0$ | The root node of $\mathbf{V}$ |
| $v$ | A node |
| $\mathbf{0}$ | The coordinate origin |
| $\mathbb{I}[\cdot]$ | The indicator function, 1 or 0 |
| $f_s(\cdot)$ | Normalization function based on $Sigmod$ |
| $f_a(v)$ | The parent node of node $v$ |
| $s(v)$ | A set of all child nodes of node $v$ |
| $H$ | The hight of the entire hierarchical structure |
| $\mathrm{M}_r$ | Root node hierarchy metric |
| $\mathrm{M}_o$ | Coordinate origin hierarchy metric |
| $\mathrm{M}_p$ | Parent node hierarchy metric |
| $\mathrm{M}_b$ | Sibling node hierarchy metric |
| $\mathrm{M}_d$ | The graph distortion |
| $\mathrm{M}_{dd}$ | The normalized graph distortion |
| $I_B$ | Horizontal hierarchical difference |
| $I_D$ | Vertical degree distribution |

### A.1    EVALUATION METRICS

Root Node Hierarchy Metric ($\mathrm{M}_r$):

$$\mathrm{M}_r = \frac{1}{|\mathbf{V}|} \sum_{v \in \mathbf{V}} \mathbb{I}_r(v)$$

$$\mathbb{I}_r(v) = \begin{cases} 1, & v = V_0 \\ \mathbb{I}[d_{\mathcal{M}}(f_a(v), V_0) < d_{\mathcal{M}}(v, V_0)] \end{cases} \tag{1}$$

where $\mathbf{V}$ denotes the set of all nodes on a hierarchical structure, $V_0$ denotes the root node of $\mathbf{V}$, $f_a(v)$ denotes parent node of node $v$, $\mathbb{I}[\cdot]$ denotes indicator function, and $d_{\mathcal{M}}(v, V_0)$ denotes the geodesic distance between $v$ and $V_0$ on the manifold space $\mathcal{M}$.

Coordinate Origin Hierarchy Metric ($\mathrm{M}_o$):

$$\mathrm{M}_o = \frac{1}{|\mathbf{V}|} \sum_{v \in \mathbf{V}} \mathbb{I}_o(v)$$

$$\mathbb{I}_o(v) = \begin{cases} 1, & v = V_0 \\ \mathbb{I}[d_{\mathcal{M}}(f_a(v), \mathbf{0}) < d_{\mathcal{M}}(v, \mathbf{0})] \end{cases} \tag{2}$$

where $\mathbf{0}$ denotes the coordinate origin.

Parent Node Hierarchy Metric ($\mathrm{M}_p$):

$$\mathrm{M}_p = \frac{1}{|\mathbf{V}|} \sum_{v \in \mathbf{V}} \mathbb{I}_p(v)$$

$$\mathbb{I}_p(v) = \begin{cases} 1, & f_a(v) = V_0 \\ \mathbb{I}[d_{\mathcal{M}}(f_a(v), v) < d_{\mathcal{M}}(f_a(f_a(v)), v)] \end{cases} \tag{3}$$

Sibling Node Hierarchy Metric ($M_b$):

$$M_b = \frac{1}{|\mathbf{V}|}(1 + \sum_{v^{l-1} \in \mathbf{V}_*} d_{rel}(v^{l-1})) \quad , l > 1$$

$$d_{rel}(v^{l-1}) = \sum_{v_i^l \in s(v^{l-1})} \frac{1}{|s_{no}(v_i^l)|} \sum_{v \in s_{no}(v_i^l)} \mathbb{I}_b(v_i^l, v)$$

$$s_{no}(v_i^l) = \{v_j^k | v_j^k \in \mathbf{V}, v_j^k \notin s(v^{l-1})$$
$$, k < l \ or \ (k = l, j < i)\}$$

$$\mathbb{I}_b(v_i^l, v) = \mathbb{I}[\max(\{d_{\mathcal{M}}(v_i^l, u) | u \in s(v^{l-1})\}) < d_{\mathcal{M}}(v_i^l, v)]$$

(4)

where $\mathbf{V}_* \subset \mathbf{V}$ denotes the set of non-leaf nodes, $v^{l-1}$ denotes the node at level $l-1$ (in the hierarchical structure) taken from $\mathbf{V}_*$ by hierarchical traversal, $d_{rel}(v^{l-1})$ denotes the proportion of distance relations satisfying Figure 2(d) that relations are between all children nodes of $v^{l-1}$ and nodes before (hierarchical traversal order) these children nodes, $v_i^l$ denotes the $i$-th node of the $l$-th level (hierarchical traversal order), $s(v^{l-1})$ denotes all child nodes of $v^{l-1}$, and $s_{no}(v_i^l)$ denotes the set of all nodes before (hierarchical traversal order) $v_i^l$.

In addition to the metrics we have proposed, we briefly introduce commonly used graph distortion metrics here. The graph distortion $M_d$ assumes that the distance $d_{\mathcal{M}}(v_i, v_j)$ between any two points should be equal to the shortest path length $d_G(v_i, v_j)$, which leads to the fact that the overall scaling up and down of node coordinate values have a significant impact on $M_d$. To overcome this problem, we divide the distance in graph distortion by the path density $d_m$ (distance per unit path length). The graph distortion $M_d$ and normalized graph distortion $M_{dd}$ are described as:

$$M_d = f_d(1), \quad M_{dd} = f_s(f_d(d_m))$$

$$f_d(x) = \frac{2}{|\mathbf{V}|(|\mathbf{V}| - 1)} \sum_{1 \leqslant i < j \leqslant |\mathbf{V}|} \left| \left( \frac{d_{\mathcal{M}}(v_i, v_j)/x}{d_G(v_i, v_j)} \right)^2 - 1 \right|$$

$$f_s(x) = 2 \cdot Sigmod(x) - 1 = \frac{2}{1 + e^{-x}} - 1$$

$$d_m = \frac{\sum_{1 \leqslant i < j \leqslant |\mathbf{V}|} d_{\mathcal{M}}(v_i, v_j)}{\sum_{1 \leqslant i < j \leqslant |\mathbf{V}|} d_G(v_i, v_j)}$$

(5)

where $f_s(x)$ denotes normalization. By dividing by $d_m$, we reduce the effect of too large or too small node coordinate values. , and make $M_d$ easier to normalize. where $d_G(v_i, v_j)$ denotes the shortest path length between $v_i$ and $v_j$.

## A.2 REDUNDANCY ANALYSIS OF EVALUATION METRICS.

We evaluate the hierarchical relationships from both horizontal and vertical perspectives: $M_b$ measures the horizontal relationship using sibling nodes, while $M_p$ assesses the vertical relationship through parent-child nodes. Additionally, since the vertical relationship is directional, we employ $M_r$ to measure the direction from the root node to the leaf nodes and $M_o$ to gauge the direction from the coordinate origin outward. In the following, we will discuss that the level of redundancy primarily depends on HRC.

Firstly, the four metrics can exhibit full redundancy. For instance, a perfectly embedded hierarchical structure would evidently achieve the highest score on all four metrics simultaneously. However, through simple illustrations, we can prove situations where there is minimal redundancy among these metrics:

- Comparing $M_b$ with the other three metrics ($M_r$/$M_o$/$M_p$): Suppose there exists a perfectly embedded complete binary tree (contains $|\mathbf{V}|$ nodes). We move the nodes on the same level to a single position, ensuring that nodes on each level form a straight line when connected. Then, we further distance the sibling nodes on the same level along the linear direction, such that the distance between two sibling nodes is equal to the distance from one node to the root. At this point, the $M_r$, $M_o$, and $M_p$ scores remain at their maximum value of 1, while $M_b$ drops from 1 to $1/|\mathbf{V}|$.

- Comparing $M_p$ with the other two metrics ($M_r$/$M_o$): Consider a perfectly embedded complete binary tree. If we invert all the nodes except for the root, such that the leaves are closest to the root and the child nodes of the root are furthest away, $M_p$ remains at its maximum value of 1, while $M_r$ and $M_o$ drop from 1 to $3/|\mathbf{V}|$. (only the root and its two child nodes satisfy the requirement).
- Comparing $M_r$ and $M_o$: Suppose there exists a perfectly embedded hierarchy. If we invert the entire structure so that the leaves are closest to the origin and the root is farthest away, $M_r$ remains at its maximum value of 1, while $M_o$ drops from 1 to $1/|\mathbf{V}|$.

The above discussion highlights that the degree of redundancy varies significantly among the metrics. They are not entirely interchangeable as they measure different aspects.

## A.3 HIERARCHICAL STRUCTURES

Horizontal Hierarchical Difference ($I_B$):

$$I_B = f_s\left(\frac{1}{|\mathbf{V}|} \sum_{v \in \mathbf{V}} \sqrt{D(v)}\right)$$

$$D(v) = \begin{cases} 0, & |s(v)| \in \{0, 1\} \\ \frac{\sum_{v_i \in s(v)}[h(v_i) - \sum_{v_j \in s(v)} \frac{h(v_j)}{|s(v)|}]^2}{|s(v)|}, & |s(v)| \geqslant 2 \end{cases}$$

(6)

where $h(v_i)$ denotes the subtree height of node $v_i$. $I_B = 0$ means a balanced tree, and $I_B$ is greater than 0 means the tree is more unbalanced. Through the normalization function $f_s$, the range of $I_B$ is constrained to the interval [0,1).

Vertical Degree Distribution ($I_D$):

$$I_D = Sigmod\left(D(\mathbf{d}_{\mathbf{v}_*})\left(\frac{f_{DCG}(\mathbf{d}_{\mathbf{v}_*}) - f_{DCG}(\mathbf{d}_{\mathbf{v}_*}^o)}{f_{DCG}(\mathbf{d}_{\mathbf{v}_*}^r) - f_{DCG}(\mathbf{d}_{\mathbf{v}_*}^o)} - \frac{1}{2}\right)\right)$$

$$\mathbf{d}_{\mathbf{v}_*} = \left\{\sum_{v_i^1 \in \mathbf{v}_*^1} \frac{|s(v_i^1)|}{|\mathbf{v}_*^1|}, \ldots, \sum_{v_i^{H-1} \in \mathbf{v}_*^{H-1}} \frac{|s(v_i^{H-1})|}{|\mathbf{v}_*^{H-1}|}\right\}$$

$$f_{DCG}(\mathbf{d}) = \sum_{d_i \in \mathbf{d}} \frac{2^{d_i} - 1}{\log_2(i+1)}, \quad i > 0$$

(7)

where $D(\mathbf{d}_{\mathbf{v}_*})$ denotes the variance of all elements in $\mathbf{d}_{\mathbf{v}_*}$, $\mathbf{d}_{\mathbf{v}_*}$ denotes the ordered set ($d_{\mathbf{v}_*}^i \leqslant d_{\mathbf{v}_*}^{i+1}$) of the mean degree of all non-leaf nodes at each level, $\mathbf{v}_*^l$ denotes the set of all non-leaf nodes at the $l$ level, $\mathbf{v}_*^1 = \{V_0\}$, $H$ denotes the hight of the entire hierarchical structure, $\mathbf{d}_{\mathbf{v}_*}^o$ denotes the ordered set of $\mathbf{d}_{\mathbf{v}_*}$ sequentially sorted, and $\mathbf{d}_{\mathbf{v}_*}^r$ denotes the ordered set of $\mathbf{d}_{\mathbf{v}_*}$ sorted in reverse order. $\mathbf{d}_{\mathbf{v}_*} = \mathbf{d}_{\mathbf{v}_*}^r$ means the node near the root node has the maximum degree. $\mathbf{d}_{\mathbf{v}_*} = \mathbf{d}_{\mathbf{v}_*}^o$ means the node near the leaf node has the maximum degree. Therefore, $I_D = 0.5$ means that the degree of nodes is evenly distributed, $I_D < 0.5$ means that the degree of nodes nearer to the leaf node is larger, and $I_D > 0.5$ means that the degree of nodes nearer to the root node is larger.

We can generate hierarchical structures with different $I_B$ and $I_D$ by two formulas, respectively.

**Control the Change of $I_B$** Recursive formula for the probability $P_r(v_i)$ of whether a node $v_i$ has children:

$$P_r(v_i) = \begin{cases} P_r(v), & |s(v)| = 1 \\ [\alpha_r + (1 - \alpha_r)\left(\frac{|s(v)|-i}{|s(v)|-1}\right)^{\alpha_t}] \cdot P_r(v), & |s(v)| \geqslant 2 \end{cases}$$

$$P_r(V_0) = 1$$

(8)

where $v_i \in s(v), v \in \mathbf{V}, \alpha_r \in [0, 1]$, and $\alpha_t > 0$. A smaller parameter $\alpha_r$ or a larger parameter $\alpha_t$ means that the generated hierarchical structure is more unbalanced ($I_B \to 1$).

**Control the Change of** $I_D$ The formula for the number $N_g(v)$ of child nodes generated by a node $v$ (calculated sequentially by hierarchical traversal):

$$N_g(v) = \min(\lfloor \mathcal{N}(\mu_v, \sigma_v^2) + 0.5 \rfloor, |\mathbf{V}| - |\mathbf{V}'|, 1)$$

$$\mu_v = \beta_{\mu_s} + (\beta_{\mu_e} - \beta_{\mu_s})\Big(\frac{|\mathbf{V}'| - 1}{|\mathbf{V}| - 1}\Big)^{\beta_{t_\mu}} \tag{9}$$

$$\sigma_v = \beta_{\sigma_s} + (\beta_{\sigma_e} - \beta_{\sigma_s})\Big(\frac{|\mathbf{V}'| - 1}{|\mathbf{V}| - 1}\Big)^{\beta_{t_\sigma}}$$

where $\lfloor \cdot \rfloor$ denotes rounded down, $\mathcal{N}$ denotes the normal distribution, $|\mathbf{V}|$ denotes the number of nodes that are scheduled to be generated, $|\mathbf{V}'|$ denotes the number of nodes that have been generated, $v \in \mathbf{V}, \beta_{t_\sigma} > 0, \beta_{t_\mu} > 0, \beta_{\mu_s} > 0, \beta_{\mu_e} > 0, \beta_{\sigma_s} \geqslant 0$, and $\beta_{\sigma_e} \geqslant 0$. The parameters $\beta_{\mu_s}$ and $\beta_{\sigma_s}$ denote the mean and standard deviation of the number of child nodes generated by the root node, respectively. The parameters $\beta_{\mu_e}$ and $\beta_{\sigma_e}$ denote the mean and standard deviation of the number of child nodes generated by the last non-leaf node in the hierarchical traversal, respectively. The more $\beta_{\mu_s}$ is greater than $\beta_{\mu_e}$, the more $I_D$ is greater than 0.5.

We also need to mix the different hierarchies to obtain a graph structure. We do this by randomly overlapping the nodes of the different hierarchies. For all nodes $\mathbf{V_i}$ and $\mathbf{V_j}$ of both hierarchies, the number of nodes to be overlapped is:

$$f_o(\mathbf{V_i}, \mathbf{V_j}) = \lfloor \mathcal{N}(\gamma_\mu, \gamma_\sigma^2) \cdot \min(|\mathbf{V_i}|, |\mathbf{V_j}|) \rfloor \tag{10}$$

where the parameters $\gamma_\mu$ and $\gamma_\sigma$ denote the mean and standard deviation of the proportion of overlapping nodes.

Although the above generation methods involve a large number of parameters, we only need to focus on the final $I_B$ and $I_D$ to analyze the different hierarchical structures. For the node classification task, we also need to generate the classification labels of the nodes. A reasonable approach is that nodes of the same class are on the same branch, and the number of nodes of different classes is the same. This will build a dataset that is easy to classify and more suitable for analyzing the HRC of the node classification task.

We show the visualization of five typical hierarchical structures in Figure 10, and their detailed parameters are described in Table 2. Each of these five typical hierarchical structures has the following characteristics: Tree1, complete n-ary tree ($I_B = 0, I_D = 0.5$); Tree2, unbalanced tree ($I_B \gg 0, I_D \approx 0.5$); Tree3, the degree of leaf nodes is greater ($I_B \approx 0, I_D \ll 0.5$); Tree4, the degree of root node is greater ($I_B \approx 0, I_D \gg 0.5$); Tree5, mixing of multiple trees. We find the parameters to generate these hierarchical structures by the hill-climbing method.

Table 2: Description of the five typical hierarchical structures and the parameters used to generate them, where $nc$ denotes the number of classes.

| Tree | $|\mathbf{V}|$ | $H$ | $I_B$ | $I_D$ | $nc$ | $\alpha_r$ | $\alpha_t$ | $\beta_{\mu_s}$ | $\beta_{\mu_e}$ | $\beta_{t_\mu}$ | $\beta_{\sigma_s}$ | $\beta_{\sigma_e}$ | $\beta_{t_\sigma}$ | Describe |
|------|------|-----|-------|-------|------|-----------|-----------|---------|---------|---------|----------|----------|----------|----------|
| Tree1 | 3280 | 8 | 0 | 0.5 | 6 | 1 | 1 | 3 | 3 | 1 | 0 | 0 | 1 | Complete 3-ary tree |
| Tree2 | 3280 | 76 | 0.2181 | 0.5016 | 6 | 0.2 | 2 | 5 | 5 | 1 | 0 | 0 | 1 | Unbalanced tree |
| Tree3 | 3280 | 11 | 0.0004 | 0.3271 | 6 | 1 | 1 | 2 | 7 | 1 | 0.4 | 1.5 | 1 | Degree of leaf is greater |
| Tree4 | 3280 | 8 | 0.0083 | 0.7791 | 6 | 1 | 1 | 6 | 1 | 0.3 | 0.1 | 1 | 1 | Degree of root is greater |
| Tree5 | 7565 | | | composed of {T1,T2,T3,T4,T5,T6,T7,T8}, $\gamma_\mu = 0.1, \gamma_\sigma = 0.1$ | | | | | | | | | | Multiple trees mixed |
| T1 | 1093 | 7 | 0 | 0.5 | - | 1 | 1 | 3 | 3 | 1 | 0 | 0 | 1 | Complete 3-ary tree |
| T2 | 1093 | 30 | 0.1095 | 0.5007 | - | 0.2 | 2 | 5 | 5 | 1 | 0 | 0 | 1 | Unbalanced tree |
| T3 | 1093 | 9 | 0.0010 | 0.3684 | - | 1 | 1 | 2 | 7 | 1 | 0.4 | 1.5 | 1 | Degree of leaf is greater |
| T4 | 1093 | 7 | 0.0099 | 0.7714 | - | 1 | 1 | 6 | 1 | 0.3 | 0.1 | 1 | 1 | Degree of root is greater |
| T5 | 1093 | 7 | 0 | 0.5 | - | 1 | 1 | 3 | 3 | 1 | 0 | 0 | 1 | Complete 3-ary tree |
| T6 | 1093 | 34 | 0.1367 | 0.5036 | - | 0.2 | 2 | 5 | 5 | 1 | 0 | 0 | 1 | Unbalanced tree |
| T7 | 1093 | 9 | 0.0009 | 0.3698 | - | 1 | 1 | 2 | 7 | 1 | 0.4 | 1.5 | 1 | Degree of leaf is greater |
| T8 | 1093 | 7 | 0.0090 | 0.7700 | - | 1 | 1 | 6 | 1 | 0.3 | 0.1 | 1 | 1 | Degree of root is greater |

## A.4 EVALUATION PROCESS

To analyze the effect of different optimization objectives and hierarchical structures on the HRC, we use five steps to construct the evaluation process of the benchmark. As shown in Figure 11, the five steps are: (1) construct the dataset based on the hierarchical structures in Section 3.2; (2) construct

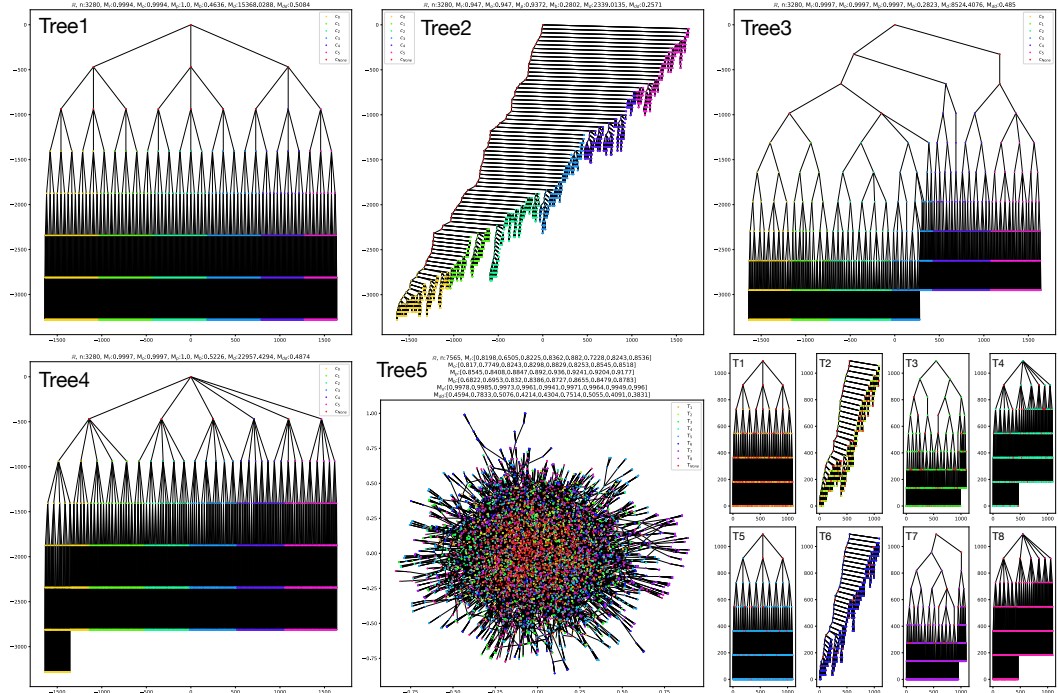

Figure 10: Five typical hierarchical structures, where n denotes the number of nodes, $c_0$-$c_5$ denote the class of nodes, $c_{None}$ denotes nodes without class, T1-T8 denote the subtrees that constitute Tress5, and $T_{None}$ denotes the overlapping nodes of different subtrees. Since the coordinates of the nodes are obtained during the visualization process, the evaluation metrics can work. $M_d$ and $M_{dd}$ represent graph distortion and normalized graph distortion, respectively.

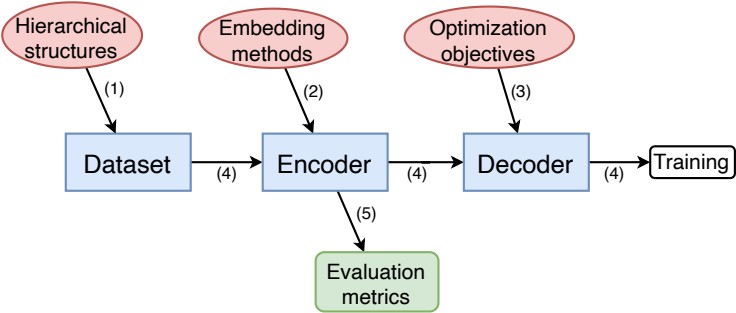

Figure 11: The evaluation process of the HRCB.

the encoder based on the embedding methods in Table 5; (3) construct the decoder based on the optimization objectives in Table 6; (4) train the encoder-decoder model; (5) evaluate the embedding of encoder outputs based on the evaluation metrics in Section 3.1.

We use the control variables method for the first and third steps to analyze the two influencing factors. The other hierarchical structures and optimization objectives can also be implemented by replacing the dataset and decoder in the first and third steps. In addition, we can also change the encoder in the second step to achieve the analysis of different embedding methods.

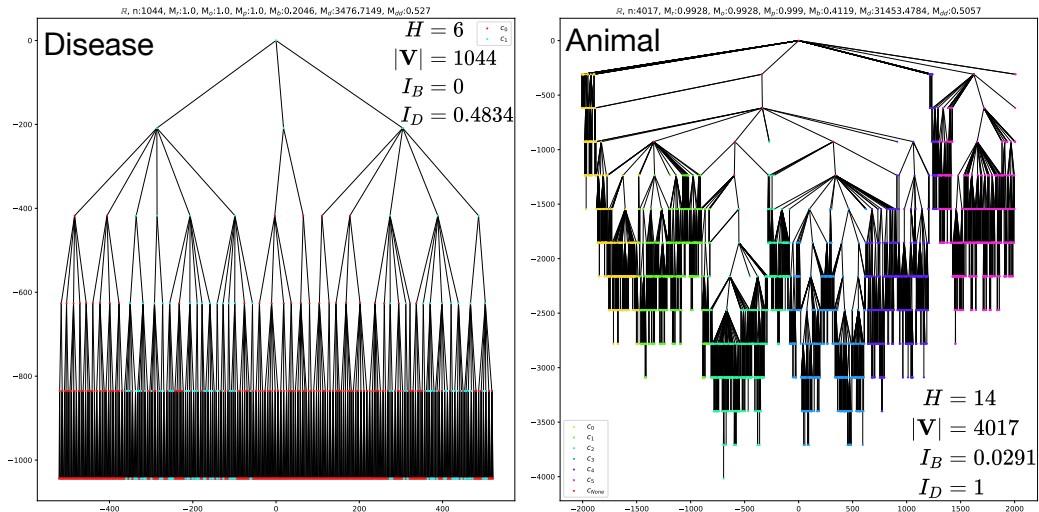

Figure 12: Visualization and description of two public hierarchical datasets, similar to Figure 10.

# B IMPLEMENTATION DETAILS

In this section, we provide a more detailed exposition of the experimental setup presented in Section 5.1.

## B.1 DATASETS

We show visualization and description of two public hierarchical datasets in Figure 12. The six classes of nodes in Animal are obtained by the method in Appendix A.3. We chose these two public hierarchical datasets because real-world graph structures often lack a standardized hierarchical relationship for us to measure HRC against. Table 3 summarizes the all datasets we utilized.

Table 3: Statistics of all datasets. Among them, Disease and Animal are utilized for the analysis of optimization objectives and embedding methods (Figures 5, 6, and 9). Trees-36, Tree5, and T1-T8 are used for the analysis related to hierarchical structures (Figures 7, 8). Trees-36 comprises 36 datasets, each with 3280 nodes and 3279 edges, with $I_B$ and $I_D$ ranging between 0 to 1.

| Name | Nodes | Edges | $I_B$ | $I_D$ | Classes |
|---|---|---|---|---|---|
| Disease | 1044 | 1043 | 0 | 0.4834 | 2 |
| Animal | 4017 | 4016 | 0.0291 | 1 | 6 |
| Trees-36 | 3280 | 3279 | (0,1) | (0,1) | - |
| Tree5 | 7565 | 8736 | - | - | - |
| T1 | 1093 | 1092 | 0 | 0.5 | - |
| T2 | 1093 | 1092 | 0.1095 | 0.5007 | - |
| T3 | 1093 | 1092 | 0.0010 | 0.3684 | - |
| T4 | 1093 | 1092 | 0.0099 | 0.7714 | - |
| T5 | 1093 | 1092 | 0 | 0.5 | - |
| T6 | 1093 | 1092 | 0.1367 | 0.5036 | - |
| T7 | 1093 | 1092 | 0.0009 | 0.3698 | - |
| T8 | 1093 | 1092 | 0.0090 | 0.7700 | - |

## B.2 MANIFOLD SPACES

The visualization of these three manifold spaces is shown in Figure 13. As illustrated in Table 4, the differences between various manifold spaces in HNNs mainly manifest in the arithmetic operations.

Table 4: Main operations on different manifold spaces. Where $\oplus$ and $\otimes$ denote addition and multiplication in manifold spaces respectively, $c$ denotes the absolute value of Riemann curvature, $K = 1/c$, $\langle \mathbf{x}, \mathbf{x} \rangle_{\mathcal{L}} = \langle \mathbf{x}, \mathbf{x} \rangle - 2x_0^2$, $||\mathbf{v}||_2 = \sqrt{\langle \mathbf{v}, \mathbf{v} \rangle_{\mathcal{L}}}$, $\mathbf{x0} := [K, 0, ..., 0] \in \mathbb{H}^{d,K}$, $\mathbf{0} := [0, ..., 0] \in \mathbb{R}^d$, $\mathbf{W}$ is European matrix, and $d_{\mathcal{M}}$ denotes geodesic distance. $P_{\mathbf{x} \to \mathbf{y}}^K(\mathbf{v})$ denotes the position of a point $\mathbf{v}$ in the tangent space of hyperbolic point $\mathbf{y}$ after translating the tangent space of hyperbolic point $\mathbf{x}$ to that of hyperbolic point $\mathbf{y}$. $\exp_{\mathbf{x}}^K(\mathbf{v})$ maps a point $\mathbf{v}$ in the tangent space of hyperbolic point $\mathbf{x}$ back to a hyperbolic point in the hyperbolic space. $\log_{\mathbf{x}}^K(\mathbf{y})$ maps a hyperbolic point $\mathbf{y}$ to its position in the tangent space of hyperbolic point $\mathbf{x}$.

| Operations | $\mathbb{R}^d$ | $\mathbb{D}^{d,K}$ $\{\mathbf{x} := [x_1, ..., x_d] \in \mathbb{R}^d : ||\mathbf{x}||_2^2 < K, x_0 = 0\}$ | $\mathbb{H}^{d,K}$ $\{\mathbf{x} := [x_0, ..., x_d] \in \mathbb{R}^{d+1} : \langle \mathbf{x}, \mathbf{x} \rangle_{\mathcal{L}} = -K, x_0 > 0\}$ |
|---|---|---|---|
| $\mathbf{x} \oplus_c \mathbf{y}$ | $\mathbf{x} + \mathbf{y}$ | $\frac{(1+2c\langle \mathbf{x},\mathbf{y}\rangle+c||\mathbf{y}||_2^2)\mathbf{x}+(1-c||\mathbf{x}||_2^2)\mathbf{y}}{1+2c\langle \mathbf{x},\mathbf{y}\rangle+c^2||\mathbf{x}||_2^2||\mathbf{y}||_2^2}$ | $\exp_{\mathbf{x}}^K(P_{\mathbf{x0}\to\mathbf{x}}^K(\log_{\mathbf{x0}}^K(\mathbf{y})))$, $\mathbf{x} \neq \mathbf{x0}, \mathbf{y} \neq \mathbf{x0}$ |
| $\mathbf{W} \otimes_c \mathbf{x}$ | $\mathbf{W}\mathbf{x}$ | $\frac{1}{\sqrt{c}}\tanh(\frac{||\mathbf{W}\mathbf{x}||_2}{||\mathbf{x}||_2}\text{arctanh}(\sqrt{c}||\mathbf{x}||_2))\frac{\mathbf{W}\mathbf{x}}{||\mathbf{W}\mathbf{x}||_2}$, $\mathbf{x} \neq \mathbf{0}$ | $\exp_{\mathbf{x0}}^K(\mathbf{W}\log_{\mathbf{x0}}^K(\mathbf{x}))$, $\mathbf{x} \neq \mathbf{x0}$ |
| $P_{\mathbf{x} \to \mathbf{y}}^K(\mathbf{v})$ | $\mathbf{v}$ | $\log_{\mathbf{y}}^K(\mathbf{y} \oplus_c \exp_{\mathbf{x}}^K(\mathbf{v}))$, $\mathbf{v} \neq \mathbf{0}$ | $\mathbf{v} - \frac{\langle \log_{\mathbf{x}}^K \mathbf{y}, \mathbf{v}\rangle_{\mathcal{L}}(\log_{\mathbf{x}}^K \mathbf{y}+\log_{\mathbf{y}}^K \mathbf{x})}{(\sqrt{K}\text{arcosh}(-c\langle \mathbf{x},\mathbf{y}\rangle_{\mathcal{L}}))^2}$, $\mathbf{x} \neq \mathbf{y}, \mathbf{v} \neq \mathbf{0}$ |
| $\exp_{\mathbf{x}}^K(\mathbf{v})$ | $\mathbf{x} + \mathbf{v}$ | $\mathbf{x} \oplus_c (\tanh(\frac{\sqrt{c}}{1-c||\mathbf{x}||_2^2}||\mathbf{v}||_2)\frac{\mathbf{v}}{\sqrt{c}||\mathbf{v}||_2})$, $\mathbf{v} \neq \mathbf{0}$ | $\cosh(\frac{||\mathbf{v}||_{\mathcal{L}}}{\sqrt{K}})\mathbf{x} + \sqrt{K}\sinh(\frac{||\mathbf{v}||_{\mathcal{L}}}{\sqrt{K}})\frac{\mathbf{v}}{||\mathbf{v}||_{\mathcal{L}}}$, $\mathbf{v} \neq \mathbf{0}$ |
| $\log_{\mathbf{x}}^K(\mathbf{y})$ | $\mathbf{y} - \mathbf{x}$ | $\frac{1-c||\mathbf{x}||_2^2}{\sqrt{c}}\text{arctanh}(\sqrt{c}|| - \mathbf{x} \oplus_c \mathbf{y}||_2)\frac{-\mathbf{x}\oplus_c\mathbf{y}}{||-\mathbf{x}\oplus_c\mathbf{y}||_2}$, $\mathbf{x} \neq \mathbf{y}$ | $\sqrt{K}\text{arcosh}(-c\langle \mathbf{x},\mathbf{y}\rangle_{\mathcal{L}})\frac{\mathbf{y}+c\langle \mathbf{x},\mathbf{y}\rangle_{\mathcal{L}}\mathbf{x}}{||\mathbf{y}+c\langle \mathbf{x},\mathbf{y}\rangle_{\mathcal{L}}\mathbf{x}||_{\mathcal{L}}}$, $\mathbf{x} \neq \mathbf{y}$ |
| $d_{\mathcal{M}}(\mathbf{x}, \mathbf{y})$ | $||\mathbf{x} - \mathbf{y}||_2$ | $(1/\sqrt{c})\text{arcosh}(1 + 2\frac{c||\mathbf{x}-\mathbf{y}||_2^2}{(1-c||\mathbf{x}||_2^2)(1-c||\mathbf{y}||_2^2)})$ | $\sqrt{K}\text{arcosh}(-c\langle \mathbf{x},\mathbf{y}\rangle_{\mathcal{L}})$ |
| $d_{\mathcal{M}}(\mathbf{X})$ | | $\left[d_{\mathcal{M}}(\mathbf{x_1}, \mathbf{X}), ..., d_{\mathcal{M}}(\mathbf{x_n}, \mathbf{X})\right]$, $d_{\mathcal{M}}(\mathbf{x_i}, \mathbf{X}) := [d_{\mathcal{M}}(\mathbf{x_i}, \mathbf{x_1}), ..., d_{\mathcal{M}}(\mathbf{x_i}, \mathbf{x_n})]^T$, $\mathbf{x_i} \in \mathbf{X}, n = |\mathbf{X}|$ | |

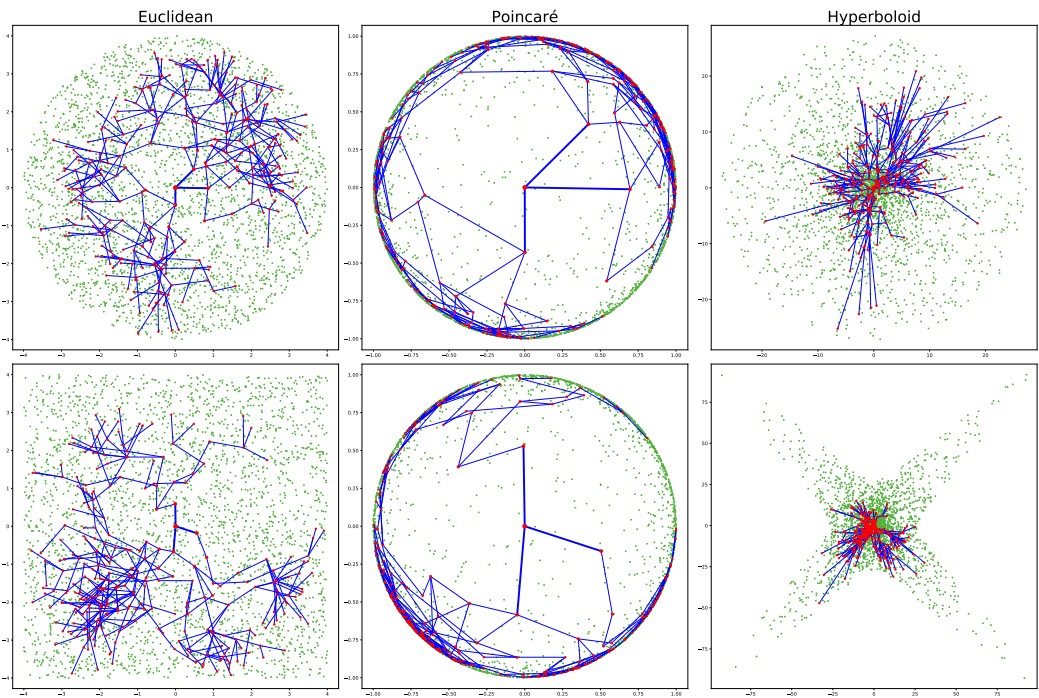

Figure 13: The same hierarchical structure is demonstrated in Euclidean space and hyperbolic space (Poincaré ball model and Hyperboloid model). We randomly generated a tree with 300 nodes (red and blue) in the circular and square Euclidean space respectively, and then obtained 3,000 points (green) by Monte Carlo sampling respectively, and finally obtained the corresponding points in the Poincaré ball and Hyperboloid models one by one by exponential mapping.

Addition and multiplication in hyperbolic space typically necessitate the use of tangent spaces, as performing calculations in these tangent spaces is more intuitive for understanding.

Table 5: Embedding Methods and their associated propagation rules. The rules govern the transition from the output $\mathbf{X}^i$ of the $i$-layer neural network to the output $\mathbf{X}^{i+1}$ of the $(i+1)$-layer neural network. The table illustrates the use of $\tilde{\mathbf{A}}$ as the normalized adjacency matrix and $\mathbf{W}^i, \mathbf{b}^i$ as the trainable parameters, along with the geodesic distance $d_{\mathcal{M}}$ (see Table 4 for details).

| Embedding Methods | Propagation rule |
|---|---|
| MLP (Ganea et al., 2018) | $\mathbf{X}^{i+1} = \sigma\Big((\mathbf{W}^i \otimes_c \mathbf{X}^i) \oplus_c \mathbf{b}^i\Big)$ |
| GCN (Liu et al., 2019) | $\mathbf{X}^{i+1} = \sigma\Big(\tilde{\mathbf{A}} \otimes_c (\mathbf{W}^i \otimes_c \mathbf{X}^i) \oplus_c \mathbf{b}^i\Big)$ |
| GAT (Zhang et al., 2019) | $\mathbf{X}^{i+1} = \sigma\Big(\mathbf{a}^i \otimes_c (\mathbf{W}^i \otimes_c \mathbf{X}^i) \oplus_c \mathbf{b}^i\Big)$ 
 $\mathbf{a}^i = Softmax(d_{\mathcal{M}}(\mathbf{W}^i \otimes_c \mathbf{X}^i)\tilde{\mathbf{A}})$ |

Table 6: Optimization Objectives and their corresponding Downstream Tasks. The objectives include Graph Distortion (GD), Hypernymy Relations (HR), Fermi-Dirac with cross-entropy (FD), and Logistic Regression (LR), while the tasks are Graph Reconstruction (GR), Link Prediction (LP), and Node Classification (NC). In this table, $d_G(u, v)$ represents the shortest path length between nodes $u$ and $v$, $\mathbf{V}$ denotes the set of all nodes, $\mathbf{E}$ indicates the set of all edges, and $L(u)$ refers to the one-hot encoding of node $u$.

| Optimization Objectives | Optimization objective function | Downstream Tasks |
|---|---|---|
| GD (Gu et al., 2019) | $\mathcal{L}_1(\Theta) = \sum_{u,v \in \mathbf{V}, u \neq v} \left\| \left(\frac{d_{\mathcal{M}}(u,v)}{d_G(u,v)}\right)^2 - 1 \right\|$ | GR |
| HR (Nickel & Kiela, 2017) | $\mathcal{L}_2(\Theta) = -\sum_{(u,v) \in \mathbf{E}} \log_e \frac{e^{-d_{\mathcal{M}}(u,v)}}{\sum_{v' \in \mathcal{N}(u)} e^{-d_{\mathcal{M}}(u,v')}}$ 
 $\mathcal{N}(u) = \{v' \mid (u,v') \notin \mathbf{E}\} \cup \{v\}, |\mathcal{N}(u)| = 11$ | GR |
| FD (Chami et al., 2019) | $\mathcal{L}_3(\Theta) = -\Big(\sum_{(u,v) \in \mathbf{E}} \log_e P(u,v)$ 
 $+ \sum_{(u',v') \notin \mathbf{E}} \log_e(1 - P(u',v'))\Big)$ 
 $P(u,v) = (e^{(d_{\mathcal{M}}(u,v)-2)} + 1)^{-1}$ | LP |
| LR (Chami et al., 2019) | $\mathcal{L}_4(\Theta) = -\sum_{u \in \mathbf{V}} \log_e \langle L(u), S(u) \rangle$ 
 $S(u) = [\frac{e^{u_1}}{\sum_{u_i \in \mathbf{u}} e^{u_i}}, \cdots, \frac{e^{u_{|\mathbf{u}|}}}{\sum_{u_i \in \mathbf{u}} e^{u_i}}]$ | NC |

### B.3 HYPERBOLIC NEURAL NETWORKS

The main differences between these network embedding models lie in their propagation rules. Table 5 summarizes the propagation rules of these three network embedding models and describes them using the general operations of manifold spaces.

### B.4 OPTIMIZATION OBJECTIVES

Table 6 displays four downstream tasks along with their corresponding optimization objectives and specific loss functions.

## C RESULT DETAILS

In Section 5.2, we conducted rigorous significance tests to analyze the relevant conclusions as to why HNNs are effective. Although this analysis is highly reliable, it combines evaluation metrics in HRCB, obscuring the performance of individual metrics. Recognizing that some researchers may be interested in different aspects of HRCB evaluation, we present the main results' performance on various metrics. Figure 14 displays the performance of the four embedding methods from Figure 5 on five metrics respectively. Figure 16 shows the performance of the four downstream task objectives

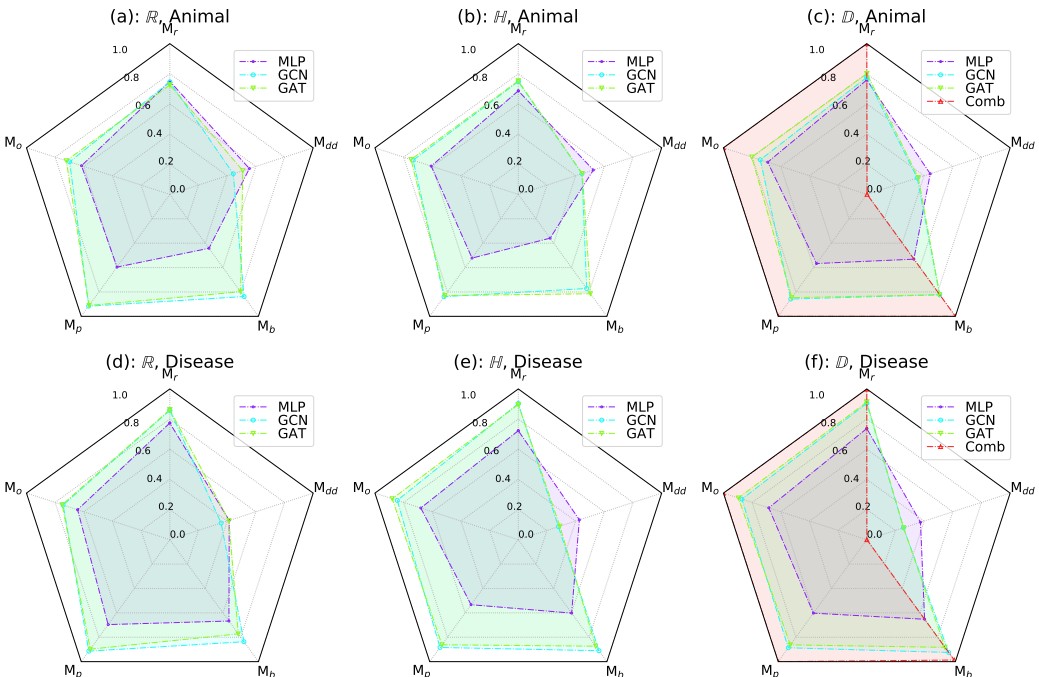

Figure 14: Results of four embedding methods (Comb,GAT,GCN,MLP) on three manifold spaces $(\mathbb{R}, \mathbb{D}, \mathbb{H})$ and two public datasets (Animal,Disease). We removed LR (NC tasks) that are not relevant to the hierarchical representation capability, so each result is an average of 24 sets of experiments ({GD,HR,FD}×eight dimensions) to obtain more general conclusions. Comb has a floating point precision of 3000 bits, while other hyperbolic neural network methods have a floating point precision of 32 bits.

from Figure 6(a) on five metrics respectively. Figure 17 illustrates the performance of all hierarchical structures from Figure 8 on five metrics respectively.

Overall, these figures are consistent with the main analysis in Section 5.2, providing additional detail on individual metrics, which we will not reiterate here. Beyond the principal analysis, we also derived some intriguing conclusions from the experimental details:

(1) The Comb method is impacted by the floating point precision. This method in Figure 14 uses 3000-bit floating point precision, but $M_b$ in Figure 14(f) still does not reach 1. This is because the vector values in hyperbolic space can be very large ($\mathbb{H}$) or very close to the boundary ($\mathbb{D}$, as shown in Figure 15).

(2) From the Figure 16, we can see that $M_{dd}$ cannot replace other evaluation metrics. The results do not show the trend that the smaller the $M_{dd}$ is, the larger the other evaluation metrics are, because the graph distortion is not designed for the hierarchical structure.

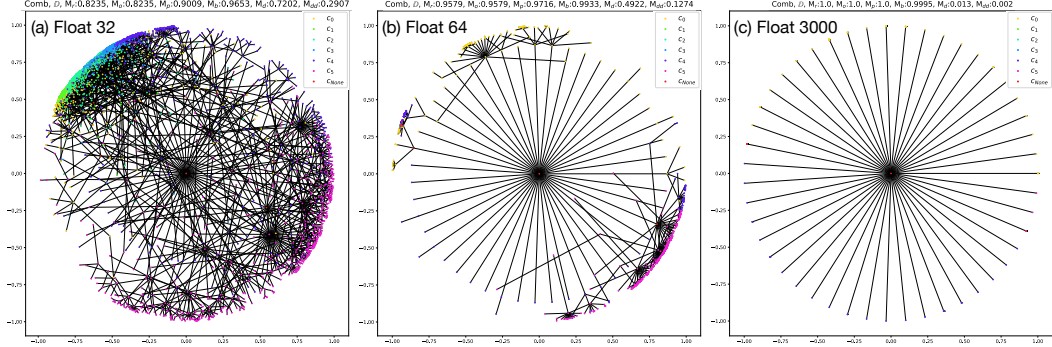

Figure 15: Visualization of Comb with different floating point precision on Animal dataset. $c_0$-$c_5$ denote the class of nodes, $c_{None}$ denotes nodes without class. It can be seen that the Comb with 32-bit floating point precision is the worst. Since the volume of the Poincaré ball model increases exponentially with the radius, the closer to the edge the better the representation is. However, the closer to the edge, the higher the floating point precision is required, otherwise it is impossible to distinguish the different nodes. In Figure (c), all the nodes except the root node are at the edge, and the different nodes too close to the edge cannot be distinguished in the visualized image.

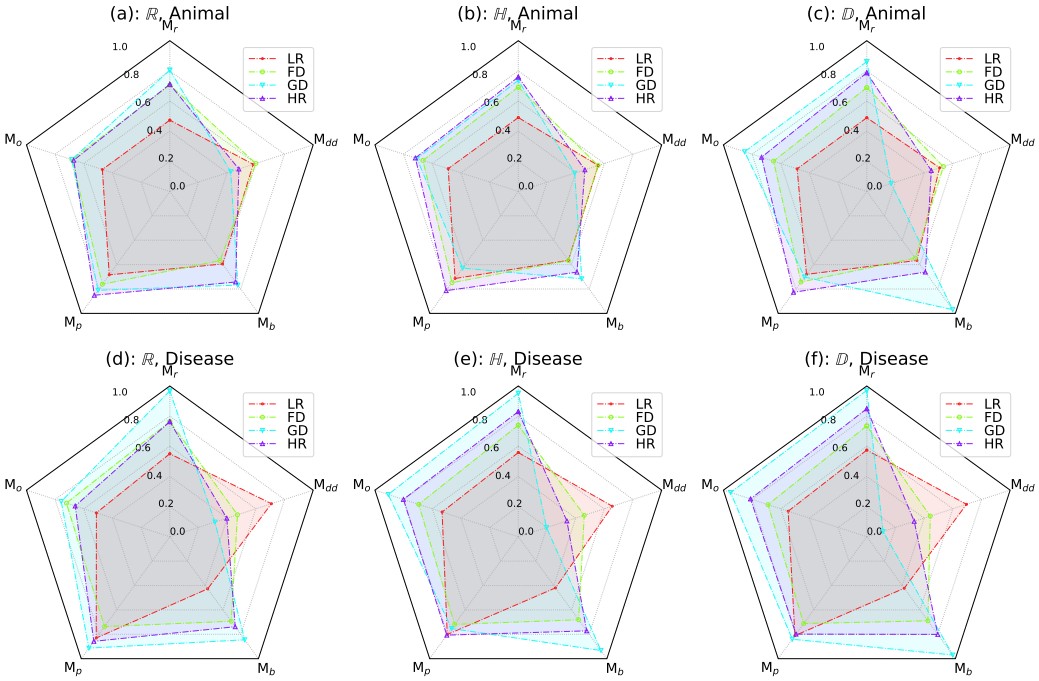

Figure 16: Results of four optimization objectives (GD,HR,FD,LR) on three manifold spaces ($\mathbb{R}, \mathbb{D}, \mathbb{H}$) and two public datasets (Animal,Disease). Each result is an average of 24 sets of experiments ({MLP,GCN,GAT}×eight dimensions) to obtain more general conclusions.

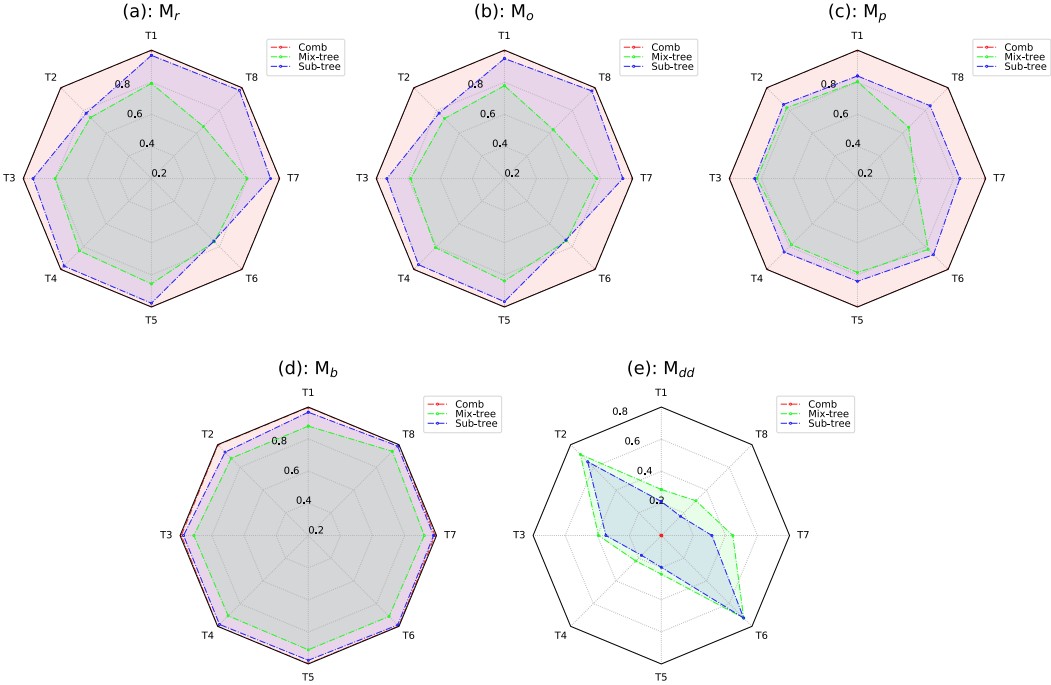

Figure 17: Mix-tree denotes training with Tree5 (mixed by T1-T8) first, and then evaluating T1-T8 separately. Sub-tree denotes that T1-T8 are trained and evaluated separately. Comb denotes that T1-T8 are trained and evaluated separately using Comb method. The results of each of the Mix-tree and Sub-tree are the average of 48 sets of experiments ($\{\mathbb{D}, \mathbb{H}\} \times \{$GD,HR,FD$\} \times$eight dimensions$\times$GCN).

