# OpenReview forum: "Why are hyperbolic neural networks effective? A study on hierarchical representation capability"
_ICLR.cc/2024/Conference — ICLR 2024 Conference Withdrawn Submission_

### Official Review · Reviewer_4wYm · 2023-10-28

**Soundness:** 3 good
**Presentation:** 1 poor
**Contribution:** 2 fair
**Rating:** 3
**Confidence:** 3

**Summary:**

The paper carefully examines the effectiveness of various Hyperbolic Neural Networks (HNNs) by measuring their Hierarchical Representation Capabilities (HRC), an evaluation process named as Hierarchical Representation Capability Benchmark (HRCB). The four metrics developed for HRCB include Root Node Hierarchy, Coordinate Origin Hierarchy, Parent Node Hierarchy, and Sibling Node Hierarchy, which altogether measure how well the hierarchical structure is embedded in hyperbolic space. The paper also proposes pre-training strategies upon improving a model's HRC, and empirically assess the relationship between HRC and downstream performance. Experimental results show that HNNs' HRC has a significant impact on downstream performance, and pre-training HNNs towards enhancing HRC can improve its performance.

**Strengths:**

- [S1] The motivation of looking back upon hyperbolic neural networks and closely assessing their effectiveness on downstream tasks with respect to theoretical work is very interesting, and I believe any insights would greatly benefit the geometric deep learning community.
- [S2] The scope of experimentation is fairly comprehensive, covering various downstream tasks, hyperbolic manifold spaces, and neural network architectures.

**Weaknesses:**

- [W1] **The overall contribution is not clear within the writing and experiments.** Is the paper hoping to show why HNNs are effective? or that HNNs do not achieve theoretically optimal embeddings?. From the experiments, it seems the goal of the paper is that 1) there exist a gap in HRC between existing HNNs and theoretically optimal embeddings and 2) reducing the gap via pre-training helps boost downstream performance, yet the overall writing (e.g., abstract and introduction) makes it confusing on what to expect from the experiments: reasons behind HNNs' effectiveness or limitations of HNNs.
- [W2] **The presented empirical observations in the text are unclear and somewhat misleading.** For instance, page 7 discusses how the LR target for NC "does not need to distinguish the position relation" among nodes, yet overfitting on LR helps improve HRC, which seems counterintuitive. Why is this the case? Furthermore, page 8 mentions how "within the applicable scope of HNNs, performance can be improved by enhancing HRC", with the node classification task being "out of scope". This is misleading considering that many previous work have shown performance boosts in node classification by leveraging hyperbolic models [A, B, C].
- [W3] **The figures showing Friedman test and Nemenyi post-hoc tests are extremely hard to read.** It would be better to categorize results based on what the authors are hoping to convey through the experiment: as an example, for Figure 6(a), if the main observation is that GD, HR, and FD help HNNs learn position relation unlike LR, it would be better to simply draw a bar chart (or multiple bar charts, one for each manifold) with targets on the x-axis and the HRC values on the y-axis. That way, we can visually observe the orderings currently written as text within the plots.
- [W4] **Downstream performance results are missing exact scores and are only compared in terms of rankings.** For Subsection 5.2.3, it would be better to simply present the downstream results in exact scores using the scoring metrics for each downstream task (F1 score for node classification, mAP for graph reconstruction and so on). This way, we can concretely estimate how much better/worse each method performed compared to another, and whether the results are within reasonable range compared to existing literature.

[A] Chami et al., Hyperbolic Graph Convolutional Neural Networks. (NeurIPS 2019)\
[B] Liu et al., Hyperbolic Graph Neural Networks. (NeurIPS 2019)\
[C] Zhu et al., Graph Geometry Interaction Learning. (NeurIPS 2020)

**Questions:**

- [Q1] The Coordinate Origin Hierarchy metric $M_o$ seems to assume that the root node is located near the origin, while this constraint is not made explicitly during training of HNNs. Considering that Combinatorial Construction [D], on the other hand, trivially satisfies this assumption, would you still consider using this metric to be valid for fair comparison?
- [Q2] Is the L strategy described in the beginning of subsection 5.2.3 equivalent to adding a weighted auxiliary loss to the downstream predictive loss? This seems very similar to how the HGCN paper used a link prediction regularization objective for node classification [A]. Thus adding a few discussion on this connection and giving the strategy a proper name rather than just "L" could help towards better clarity.

[D] Sala et al., Representation Tradeoffs for Hyperbolic Embeddings. (ICML 2018)

---

### Official Review · Reviewer_u1No · 2023-10-31

**Soundness:** 3 good
**Presentation:** 3 good
**Contribution:** 3 good
**Rating:** 5
**Confidence:** 3

**Summary:**

The paper elucidates the scope and applicability of HNNs through quantitative analysis of their HRC. It provides guidance on enhancing HNNs by identifying factors that improve their hierarchical representation capability. In particular,

- This paper proposes a benchmark (HRCB) to quantitatively evaluate the hierarchical representation capability (HRC) of HNNs. HRCB includes metrics to assess horizontal relationships (sibling nodes) and vertical relationships (parent-child nodes) in a hierarchy.

- Experiments using HRCB show that HNNs do not achieve the theoretical optimal embedding in hyperbolic space. Their HRC is significantly lower than combinatorial construction methods.

- Analysis reveals two key factors influencing HNNs' HRC: (1) Optimization objectives that help distinguish positional relationships between nodes, and (2) Training data structured as a complete n-ary tree.

- The paper proposes pre-training strategies to enhance HNNs' HRC based on these insights. Experiments show improved downstream task performance from enhanced HRC, validating the analysis.

**Strengths:**

The author made a very interesting observation and conducted numerous experiments to support their findings.

**Weaknesses:**

There are several points of concern from me:

- The HRCB metrics presented presume the root node is positioned at the highest point. Additionally, the authors assume that the root node should be close to the origin. This isn't accurate for all HNNs, as detailed in [1]. If these assumptions are invalid, the four evaluation criteria proposed might not be accurate.

- The research primarily uses two datasets (Disease and Animal) for analysis. Including a broader range and real-world datasets would provide more robust conclusions.

- The author's description of pre-training strategies isn't very clear. Could this be elaborated more?

- The comparison is made with the GCN model, but the HGCN isn't considered. This is an omission.

- While the findings are intriguing, there are various forms of HNN currently available, such as those based on the tangent space or being fully hyperbolic. The author doesn't seem to address this aspect.

[1] Menglin Yang et al. Hyperbolic Representation Learning: Revisiting and Advancing. ICML 2023.

**Questions:**

1. Could you provide more details on the data generation process for the hierarchical structures?

---

### Official Review · Reviewer_pJty · 2023-11-05

**Soundness:** 3 good
**Presentation:** 3 good
**Contribution:** 3 good
**Rating:** 3
**Confidence:** 4

**Summary:**

This paper studies a benchmark for evaluating the hierarchical representation capacity (HRC) of the hyperbolic neural network (HNN). The empirical study shows the HRC can be affected by the optimization objectvie and the training data. This observation facilitate to develop pre-training strategies to enhace the HRC of HNN, improving the learning capacity and performance of the neural network. This paper shows some interesting observations, but it lacks of the generalization of HNN to other tasks.

**Strengths:**

1-This paper first study the hierarchical representation capacity (HRC) of the hyperbolic neural network (HNN) and aim to answer how the HNN works.
2-A benchmark, including data and metric, is proposed to evaluate the HRC.
3-Three pre-training methods are proposed to improve the learning capacity and the performance.

**Weaknesses:**

1-This paper only studies the HRC in the graph dataset, the observation on text and graph data is missed.

2-The experiments on pre-training method misses formulation of the losses. In addition, why the GD as objective function can attain good performance, please explain?

**Questions:**

1-This paper study the HRC for the graph data, there is another metric, called delta-hyperbolicity, can be used to evaluate the hierarchiy of the data, I want to see does the value of delta-hyperbolic matches the value of the metrics proposed in this paper.

2-This paper only studies the HRC in the graph dataset, does the conclusion is also hold in text and image data?

3-The proposed pre-training method should also be evaluated in other datasets.